# Quantitative Analysis of *Camellia oleifera* Seed Saponins and Aqueous Two-Phase Extraction and Separation

**DOI:** 10.3390/molecules28052132

**Published:** 2023-02-24

**Authors:** Lifang Zhu, Shanshan Wang, Faling Wan, Yihong Zhou, Zongde Wang, Guorong Fan, Peng Wang, Hai Luo, Shengliang Liao, Lu He, Yuling Yang, Xiang Li, Xiuxiu Zou, Shangxing Chen, Ji Zhang

**Affiliations:** National Forestry and Grassland Bureau Woody Spice (East China) Engineering Technology Research Center, the Institute of Plant Natural Products and Forest Products Chemical Engineering, College of Forestry, Jiangxi Agricultural University, Nanchang 330045, China

**Keywords:** *Camellia oleifera* Abel. seed meal, UV spectrophotometry, methanol extract, aqueous two-phase system, liquid chromatography

## Abstract

At present, the technology used for the extraction and purification of *Camellia oleifera* saponins generally has the problems of high cost and low purity, and the quantitative detection of *Camellia oleifera* saponins also has the problems of low sensitivity and easy interference from impurities. To solve these problems, this paper aimed to use liquid chromatography for the quantitative detection of *Camellia oleifera* saponins, and to adjust and optimize the related conditions. In our study, the average recovery of *Camellia oleifera* saponins obtained was 100.42%. The RSD of precision test was 0.41%. The RSD of the repeatability test was 0.22%. The detection limit of the liquid chromatography was 0.06 mg/L, and the quantification limit was 0.2 mg/L. In order to improve the yield and purity, the *Camellia oleifera* saponins were extracted from *Camellia oleifera* Abel. seed meal by methanol extraction. Then, the extracted *Camellia oleifera* saponins were extracted with an ammonium sulfate/propanol aqueous two-phase system. We optimized the purification process of formaldehyde extraction and aqueous two-phase extraction. Under the optimal purification process, the purity of *Camellia oleifera* saponins extracted by methanol was 36.15%, and the yield was 25.24%. The purity of *Camellia oleifera* saponins obtained by aqueous two-phase extraction was 83.72%. Thus, this study can provide a reference standard for rapid and efficient detection and analysis of *Camellia oleifera* saponins for industrial extraction and purification.

## 1. Introduction

*Camellia oleifera* Abel. (*C. oleifera)*, an oil crop, is widely cultivated in China. *C. oleifera* Abel. seed meal, the residue of *C. oleifera* Abel. processing, is often used in traditional agriculture as a fertilizer, with a general utilization rate and low added value [1]. *C. oleifera* Abel. seed meal is naturally rich in *C. oleifera* saponins, which can be used as a raw material to purify and separate *C. oleifera* saponins products with high purity and high added value [2]. Therefore, *C. oleifera* Abel. seed meal is used to extract high-purity *C. oleifera* saponins, which can greatly increase the economic benefits of the *C. oleifera* industry. *C. oleifera* saponins are an oleanane-type pentacyclic triterpene saponin [3]. Their basic structure is composed of triterpene saponins, sugar bodies, and organic acids [4]. They exhibit the general properties of saponins, such as good hemolytic, antibacterial, and surface activity [5,6]. In recent years, *C. oleifera* saponins, traditionally used as emulsifiers, have been industrially utilized, including use as daily chemicals, pesticides, medicine, and food, which shows the potential for their broad application and economic value [7,8,9].

The quantitative analysis of *C. oleifera* saponins requires certain selectivity due to their structural complexity and molecular diversity [2]. Generally, the quantitative analysis of *C. oleifera* saponins are performed mainly via gravimetry, colorimetry, and high-performance liquid chromatography [10,11,12,13,14]. Although the determination results of gravimetry are relatively stable, the experimental procedure is cumbersome and time-consuming, and it requires many reagents and drugs. Therefore, the cost of the gravimetry is high [15]. Colorimetry has high sensitivity but is easily affected by impurities, such as flavonoids and phenolic compounds. Liquid chromatography is simple, rapid, accurate, and reproducible. Moreover, the literature on the detection of saponins by liquid chromatography has verified its advantages. Sun [16] et al. established a method for the simultaneous determination of five triterpenoid saponins in Clematis using high-performance liquid chromatography–evaporative light scattering, and the total recovery of each of the five analytes was between 91.3% and 99.5%. Ganzera et al. [17] separated triterpenoid saponins from purslane using high-performance liquid chromatography and determined that the total saponins content in the sample varied from 1.1 to 13.0%.

The current primary methods for the industrial extraction of *C. oleifera* saponins include aqueous extraction, organic solvent methods, and auxiliary extraction [18,19,20]. The methods that have been utilized to enhance the purity of *C. oleifera* saponins mainly include chemical precipitation, recrystallization, membrane purification, etc. [21,22]. However, these methods have their limitations. For example, although chemical precipitation is simple to operate, it leads to residual impurities. Recrystallization is time-consuming, and the membrane of the membrane separation method is easily blocked. Therefore, studies have increasingly used aqueous two-phase extraction and purification technologies on the basis of the selectivity of the extracted substance in the upper and lower phases [23,24,25]. Extraction and separation are accomplished on the basis of the differential solubility and partition coefficient of the substance in the two phases [26,27,28]. Wei [29] et al. constructed a new recyclable two-phase aqueous system for the distribution of *C. oleifera* saponins using the temperature-responsive polymer PN and the pH-responsive polymer PADB4.78. Motlagh [30] et al. used an aqueous two-phase system composed of polyethylene glycol and K_2_HPO_4_ solution to extract beetroot saponins, which greatly improved foam volume and stability.

In this study, the quantitative analyses of *C. oleifera* saponins via liquid chromatography, colorimetry, and gravimetry methods were studied and compared. High-value and high-purity *C. oleifera* saponins were prepared from *C. oleifera* seed meal. First, the *C. oleifera* saponins were extracted from *C. oleifera* seed meal via a methanol extraction method. The crude *C. oleifera* saponins were separated and purified via aqueous two-phase extraction, resulting in a high-value and high-efficiency utilization of *C. oleifera* processing residues. This has important practical significance for improving the comprehensive output of the *C. oleifera* industry and promoting the industrial utilization of *C. oleifera* saponins.

## 2. Results and Discussion

### 2.1. Quantitative Analysis of C. oleifera Saponins

#### 2.1.1. UV Spectrophotometry

A standard curve was drawn with the abscissa as the mass concentration of the *C. oleifera* saponins and the ordinate as the absorbance. The regression equation was *y* = 6.1374*x* − 0.0151, where *y* is the absorbance, and *x* is the mass concentration of the *C. oleifera* saponins (R^2^ = 0.9972). The standard curve is shown in Figure 1.

The content of saponins in the sample was detected by UV spectrophotometry, and the measured absorbance was 0.677 a.u. The mass concentration of the *C. oleifera* saponins in the sample was calculated by the standard curve in Figure 1. Finally, the content of the sample of the *C. oleifera* saponins was determined as 62.00% using Formula (2).

The detection limit of UV spectrophotometry was 0.15 mg/L, and the quantification limit was 0.49 mg/L using Formulas (3) and (4). The matrix effect was determined to be 26.24% by using Formula (7). This value was between 20% and 50%, which was a medium matrix effect. This indicated that spectrophotometry was more easily interfered with by other impurities in the solution when used to measure the content of *C. oleifera* saponins.

#### 2.1.2. Analysis of Detection Results by Liquid Chromatography

##### Standard Curve Based on Liquid Chromatography

A standard solution of 100 mg *C. oleifera* saponins was accurately weighed and dissolved in a 10 mL volumetric flask with pure water. After mixing evenly, a standard solution of 10 mg/mL *C. oleifera* saponins was obtained via ultrasonic vibration for 5 min. A volume of 5 mL of standard solution was transferred into a 10 mL volumetric flask with pure water to obtain a 5 mg/mL *C. oleifera* saponins standard solution. Standard working solutions of 2.5, 1.25, 0.625, and 0.3125 mg/mL were obtained from repeated operations. An amount of 1 mL of standard liquid was transferred via a syringe to a 0.22 µm microwell filter membrane and loaded into the sample bottle. The test results are shown in Table 1.

Because the *C. oleifera* saponins were composed of a variety of saponins monomers, the chromatogram exhibited multiple monomer peaks. The sum of the sealing areas of multiple monomers represented the peak area of *C. oleifera* saponins.

Taking the concentration of the *C. oleifera* saponins standard working solution as the abscissa, and the sum of the peak areas of the six main chromatographic peaks as the ordinate, the standard curve of *C. oleifera* saponins content was calculated, and it is shown in Figure 2.

The detection limit of the liquid chromatography limit was determined as 0.06 mg/L, and the quantification was determined as 0.2 mg/L using Formulas (5) and (6). The matrix effect was determined as 17.74% using Formula (7), and the value <20% was a weak matrix effect, which could be ignored without compensation measures.

##### Precision Test

An amount of 10.0 mg *C. oleifera* saponins was dissolved in a 10 mL volumetric flask with pure water and then sonicated for 5 min to obtain a 1 mg/mL *C. oleifera* saponin standard solution. An appropriate amount of standard solution was filtered through a 0.22 µm microporous filter membrane and then loaded into the sample injection bottle for sample injection and detection.

In accordance with the liquid chromatography method described in this study, *C. oleifera* saponins were continuously injected, six times a day. The results are shown in Table 2.

As shown in Table 2, the average peak area of the *C. oleifera* saponins was 827.83 mAU and the RSD of *C. oleifera* saponins peak area was 0.41%, indicating the precision and repeatability of the test.

##### Repeatability Test

A *C. oleifera* saponins sample (200 mg) was weighed, and the procedure was repeated according to the conditions for liquid chromatography sample preparation in this experiment. The peak of the *C. oleifera* saponins was obtained, as shown in Figure 3. Three single peaks were detected at 33, 35, and 36 min, respectively.

The sum of the chromatographic peak areas of six samples was recorded, and the results are shown in Table 3.

The relative standard deviation (RSD) result was 0.22% (*n* = 6), indicating that the method had good repeatability. The calculated average peak area of the *C. oleifera* saponins was 1708.74 mAU, and the average peak area of *C. oleifera* saponins was substituted into the standard equation of the *C. oleifera* saponins *y* = 1476.1*x* − 249.17. The calculated average mass fraction of the *C. oleifera* saponins samples was 70.08%.

##### Average Spike Recovery Test

An amount of 100.0 mg of *C. oleifera* saponins was dissolved in a 10 mL volumetric flask with pure water and then sonicated for 5 min to obtain a 10 mg/mL *C. oleifera* saponins standard solution.

A 1 mL aliquot of the *C. oleifera* saponins sample solution was treated with 0.5 mL, 1 mL, and 1.5 mL of *C. oleifera* saponins standard solutions, and the conditions for improved detection in this experiment were tested. The results are shown in Table 4.

The RSD peak area was used as an indicator of instrument precision. The RSD of the precision test was 0.41% (*n* = 6), indicating strong accuracy. The RSD of the repeatability test was 0.22% (*n* = 6), indicating good repeatability. The recoveries ranged from 97.20% to 104.50%, which met the criteria for recovery. The average recovery of the *C. oleifera* saponins obtained in this method was 100.42%.

#### 2.1.3. Analysis of Weight Test Results

The weight of the *C. oleifera* saponins sample (m_1_) was 1.50 g, and the mass of the receiving bottle after drying (m_2_) was 125.90 g. The sum of the mass of the receiving bottle and the extract was 126.23 g after the constant weight was determined. Based on the above data, the following formula was used to analyze the weight of the saponins:$$W=\frac{(126.23-125.90)\times 1223.54}{501\times 1.50}\times 100\%=53.55\%$$

In the parallel experiment, the *C. oleifera* saponins sample (m_1_) was 1.50 g, while the mass of the receiving bottle after drying (m_2_) was 133.10 g. The sum of the mass of the receiving bottle and the extract after the experimental procedure was constant (m_3_) was 133.42 g. Based on these data, the average mass fraction was calculated using the following formula:$$W=\frac{(133.42-133.10)\times 1223.54}{501\times 1.50}\times 100\%=52.24\%$$

The average mass fraction of the gravimetric method was 52.24%. The experimental results were relatively accurate, but lower than the results obtained via liquid chromatography. The operation was cumbersome and time-consuming, and the detection efficiency was poor.

### 2.2. Results of Alcoholic Extraction of C. oleifera Saponins

#### 2.2.1. The Effect of the Extracted Liquid–Solid Ratio on the Yield of *C. oleifera* Saponins

As shown in Figure 4, the yield of the *C. oleifera* saponins increased first and then decreased with the increase in the liquid–solid ratio. The maximum yield was 3.5 mL/g, and the yield of *C. oleifera* saponins was 18.35%. When the quality of raw materials reached a specific level, the addition of a methanol solution reduced the concentration of the *C. oleifera* saponins. As the difference in concentration between the *C. oleifera* saponins and solvent increased, it accelerated the speed of mass transfer. The yield of the *C. oleifera* saponins was increased. However, when the liquid–solid–liquid ratio was too large, it increased the dissolution of impurities, resulting in a decline in the yield of the *C. oleifera* saponins with the increase in the solid–liquid ratio [31]. Based on the overall analysis, a liquid–solid ratio of 3.5:1 was selected.

#### 2.2.2. The Effect of Extraction Temperature on the Yield of *C. oleifera* Saponins

As shown in Figure 5, with the increase in temperature, the yield of the *C. oleifera* saponins increased first and then decreased. When the extraction temperature was 60 °C, the yield of the *C. oleifera* saponins reached 18.14%. The amount of *C. oleifera* saponins dissolved was relatively small at low temperatures. The yield was also low. However, when the temperature was too high, the volatility of the solvent increased significantly, resulting in irreversible degeneration and solidification of impurities, such as proteins and pectin, and precipitation after combining with the *C. oleifera* saponins, which decreased the content in the solution [32]. Therefore, the most suitable temperature was 60 °C.

#### 2.2.3. The Effect of Methanol Concentration on the Yield of *C. oleifera* Saponins

As shown in Figure 6, with the increase in the methanol concentration, the yield of the *C. oleifera* saponins first slowly increased and then decreased. When the methanol concentration was 75%, the yield of the *C. oleifera* saponins reached a peak value of 18.94%. The low-concentration methanol leaching solution carried multiple water-soluble impurities and bubbles which were difficult to eliminate. Within a certain range of methanol concentration, increasing the methanol concentration increased the yield of the *C. oleifera* saponins. However, if the methanol concentration was too high, the solubility of the *C. oleifera* saponins decreased and the extraction efficiency was reduced [33]. Therefore, based on the overall analysis, a 75% methanol concentration was selected.

#### 2.2.4. The Effect of Extraction Time on the Yield of *C. oleifera* Saponins

As shown in Figure 7, the content of the *C. oleifera* saponins first increased and then decreased with the increase in extraction time. When the extraction time was 180 min, the yield of the *C. oleifera* saponins was 19.73%, reaching a maximum value. When the extraction time was short, the methanol and *C. oleifera* Abel. seed meal failed to mix completely, and the methanol failed to completely dissolve the *C. oleifera* saponins. With the increase in extraction time, the dissolution of impurities increased and the yield of the *C. oleifera* saponins decreased. Therefore, the most suitable extraction time was 180 min.

#### 2.2.5. Orthogonal Test and Analysis of *C. oleifera* Saponins Extraction

Based on the orthogonal test of L9 (3^4^), four factors and value intervals were obtained. The experimental results are shown in Table 5.

As shown in Table 5, the impact of the four factors on yield was in the order of D > C > A > B, suggesting that the effect of the liquid–solid ratio was greater than that of the methanol concentration, which was greater than that of the extraction temperature, which was greater than the effect of the extraction time. Thus, the liquid–solid ratio and methanol concentration had the greatest impact on the yield of the *C. oleifera* saponins extraction. The liquid–solid ratio and methanol concentration were significant factors affecting the extraction yield of the *C. oleifera* saponins. The optimum extraction conditions of the *C. oleifera* saponins were as follows: the extraction temperature was 55 °C, the extraction time was 210 min, the methanol concentration was 75%, and the liquid–solid ratio was 4:1. Under these conditions, multiple experiments were carried out. The average yield of the obtained *C. oleifera* saponins was 25.26%, and the average purity of the *C. oleifera* saponins was 36.15%. Bao [7] et al. found that the yield of the *C. oleifera* saponins was 14.71% by ultrasonic-assisted ethanol extraction. In contrast, the yield of the *C. oleifera* saponins obtained by our methanol extraction method was significantly improved. The reason might be that the polarity of the methanol was greater and our experimental conditions were better, and the varieties of *C. oleifera* were also better, containing more *C. oleifera* saponins, thus, the extraction effect was better.

### 2.3. Analysis of C. oleifera Saponins via Aqueous Two-Phase Extraction Results

#### 2.3.1. Effect of Extraction Temperature on Two-Phase Extraction and Purification of *C. oleifera* Saponins

As shown in Figure 8, the purity of the *C. oleifera* saponins increased first and then decreased with the increase in extraction temperature. When the extraction temperature reached 30 °C, the purity reached the highest value of 81.12%. A further increase in the extraction temperature sharply decreased the purity. An increase in temperature led to an increase in the solubility of the *C. oleifera* saponins in the aqueous two-phase system. However, excessive temperatures led to the denaturation of impurities, such as proteins and polysaccharides, and precipitation via encapsulation of the *C. oleifera* saponins, resulting in reduced levels of purity [34]. Therefore, the most suitable extraction temperature was 30 °C.

#### 2.3.2. The Effect of Propanol Mass Fraction on *C. oleifera* Saponins Aqueous Two-Phase Extraction and Purification

As shown in Figure 9, the purity of the *C. oleifera* saponins increased first and then decreased with the increase in the mass fraction of propanol. When the mass fraction of propanol was 11%, the purity of the *C. oleifera* saponins reached its peak value of 78.06%. However, when the mass fraction of propanol was too high, the purity of the *C. oleifera* saponins showed a downward trend. With the increase in the mass fraction of propanol, the ability of the upper phase to absorb the *C. oleifera* saponins increased, which increased the solubility of the *C. oleifera* saponins in the upper phase. However, it led to protein denaturation, resulting in the partial encapsulation of the *C. oleifera* saponins precipitated into the lower phase. Therefore, the mass fraction of propanol was selected as 11%.

#### 2.3.3. Effect of Ammonium Sulfate Mass Fraction on *C. oleifera* Saponins Aqueous Two-Phase Extraction and Purification

As shown in Figure 10, with the increase in the mass fraction of ammonium sulfate, the purity of the *C. oleifera* saponins first increased and then decreased. When the mass fraction of ammonium sulfate was 10%, the purity of the *C. oleifera* saponins reached the maximum value of 80.85%. With the increase in the ammonium sulfate mass fraction, the solubility of the *C. oleifera* saponins in the upper phase increased. However, an excessive ammonium sulfate mass fraction led to salting out, and the *C. oleifera* saponins were precipitated to the lower phase. At this time, the *C. oleifera* saponins content was reduced. Therefore, it was most appropriate to select a 10% mass fraction of ammonium sulfate.

#### 2.3.4. Orthogonal Test and Analysis of *C. oleifera* Saponins Aqueous Two-Phase Extraction and Purification

Based on orthogonal test on L9 (3^3^), three influencing factors and value intervals were obtained. The experimental results are shown in Table 6.

As shown in Table 6, the effect of the three factors on the purity of the *C. oleifera* saponins was in the order of C > B > A. That is, the effect of the mass fraction of propanol was greater than the effect of the mass fraction of ammonium sulfate, and the effect of the mass fraction of ammonium sulfate was greater than that of the extraction temperature of the *C. oleifera* saponins. The mass fractions of propanol and ammonium sulfate were significant factors. According to the analysis and experimental results, the optimum conditions for two-phase extraction and purification of the *C. oleifera* saponins was an extraction temperature of 20 °C, a 9% mass fraction of ammonium sulfate, and an 11% mass fraction of propanol. Under these experimental conditions, the yield of the *C. oleifera* saponins obtained was 61.34%, and the average purity was 83.72%. Compared with the purity of 75.79% ginsenoside obtained by enzymatic hydrolysis from Han [20] et al. in the aqueous system of low eutectic solvo salt, higher purity *C. oleifera* saponins could be obtained by the propanol/ammonium sulfate system. This may be mainly due to the high selectivity of the propyl alcohol/(NH_4_) _2_SO_4_ system to the *C. oleifera* saponins in the aqueous two-phase system. The ions that could be used to precipitate *C. oleifera* saponins include Ba^2+^ and Ca^2+^ ions. These ions could be combined with the carboxyl groups in the *C. oleifera* saponins to obtain higher purity *C. oleifera* saponins.

## 3. Materials and Methods

### 3.1. Materials

*C. oleifera* Abel. seed meal (variety: Changlin 3) was provided by Jiangxi Zhongye Tea Technology Co., Ltd (Jiangxi, China). (Approval Number SC10236112411503).

Raw material treatment was as follows: *C. oleifera* Abel. seed meal was pulverized (Disintegrator, 103B, Rui’an Yongli Pharmaceutical Machinery Co., Ltd., Rui’an, China), and dried by passing it through a 60-mesh screen (GB/T6003, Shaoxing Jinhang Instrument Co., Ltd., Shaoxing, China), followed by decreasing and drying with petroleum ether, with a moisture content of < 6% [35].

*C. oleifera* saponins (purity, ≥98%) were purchased from Solarbio Corporation (Beijing, China). Anhydrous methanol, concentrated sulfuric acid, petroleum ether (60–90), ammonium bicarbonate, acetone, phosphoric acid, and sodium hydroxide were purchased from Xilong Science Co., Ltd (Shantou, China). Vanillin, N-Propanol, and calcium oxide were purchased from Tianjin Damao Chemical Reagent Factory (Tianjin, China). Ammonium sulphate was purchased from Daduhe Road Company (Chengdu, China). All other chemicals were of analytical grade except ammonium sulfate and hydrochloric acid.

### 3.2. Experimental Method

#### 3.2.1. Quantitative Analysis of *C. oleifera* Saponins

##### UV Spectrophotometry

Preparation of *C. oleifera* saponins standard solution and sample solutions

We accurately weighed 100.0 mg of 98% *C. oleifera* saponins standard, dissolved it in 20% ethanol solution, and diluted it in a 100 mL volumetric flask to obtain a standard solution of 1 mg/mL. The sample solution (1 mg/mL) was prepared similarly to the standard solution described above.

2.Selection of the maximum absorption wavelength

Using a pipette gun (100–1000 μL/1–10 mL Shanghai Baoyude Scientific Instrument Co., Ltd., Shanghai, China), a 0.5 mL standard solution and 0.5 mL sample solution were added to a test tube. It was then absorbed with 0.8% vanillin and 0.5 mL of extract–ethanol solution (0.80 g vanillin dissolved in 10 mL anhydrous ethanol). The solution was shaken well and then soaked in an ice-cold water bath for 10 min, followed by the addition of 4 mL of 77% concentrated sulfuric acid [36]. The mixture was heated for 15 min in a constant temperature water bath at 60 °C (DF-101S, Zhengzhou Dufu Instrument Factory, Zhengzhou, China), followed by immersion in an ice water bath for 10 min. The solution was then left at room temperature for recovery and removal [36]. The maximum absorption wavelength was determined by a UV spectrophotometer (TU-1950, Beijing Puxi General Instrument Co., Ltd., Beijing China) in the range of 200–800 nm, with a 20% ethanol solution as a blank control [37]. Finally, the maximum absorption wavelength was determined to be 461 nm.

3.Drawing of standard curve

Aliquots of 0.1 mL, 0.2 mL, 0.3 mL, 0.4 mL, 0.5 mL, 0.6 mL, 0.7 mL, and 0.8 mL of the standard solution were added to the labeled test tubes, followed by mixing with 0.9 mL, 0.8 mL, 0.7 mL, 0.6 mL, 0.5 mL, 0.4 mL, 0.3 mL, and 0.2 mL of 20% ethanol solution, respectively. The reagent blank was treated with 1 mL of 20% ethanol solution, followed by the addition of 0.5 mL of 8% vanillin–ethanol solution. The test tube was then left in an ice water bath for 10 min, followed by the addition of 4 mL of 77% concentrated sulfuric acid. It was then heated in a water bath at a constant temperature of 6 °C for 15 min. It was immersed in an ice water bath for 10 min and transferred to a room temperature environment [38]. Then, the blank reagent was used to measure the absorbance of other groups and the data were recorded, and three parallel experiments were performed. A standard curve was drawn based on the data. The absorbance of the test samples was determined, and the concentration and content of *C. oleifera* saponins were calculated.

4.Calculation method

(1)Calculation of yield


(1)
$$D=\frac{n}{N}\;\times \;100\%$$


In the formula above, *n* is the total mass of crude *C. oleifera* saponins after drying (g), *N* denotes the total mass of *C. oleifera* Abel. seed meal (g), and *D* is the yield (%).

(2)Calculation of purity


(2)
$$W=\frac{C\;\times \;5.5\;\times \;10}{m}\;\times \;100\%$$


In the formula, *C* is the mass concentration of the *C. oleifera* saponins after dilution based on ultraviolet spectrophotometry. It was obtained by measuring the absorbance, and then converted according to the standard curve shown in Figure 2. The symbol *m* is the mass of the *C. oleifera* saponins (mg), while *W* is the purity of the *C. oleifera* saponins (%).

(3)Limit of detection and limit of quantitation of UV spectrophotometry


(3)
$$D_{L}=\frac{k_{1}\times S}{b}\;$$



(4)
$$Q_{L}=\frac{k_{2}\times S}{b}\;$$


In the formula, *D_L_* is the limit of detection, *Q_L_* is the limit of quantitation, *k* (*k*_1_ = 3, *k*_2_ = 10) is the confidence factor, *S* is the blank standard deviation, and *b* is the slope of the standard curve in the low concentration range.

(4)Limit of detection and limit of quantitation of liquid chromatography


(5)
$$D_{L}=\frac{k_{1}\times N_{d}\times C}{H}\;$$



(6)
$$Q_{L}=\frac{k_{2}\times N_{d}\times C}{H}\;$$


In the formula, *D_L_* is the limit of detection, *Q_L_* is the limit of quantitation, *k* (*k*_1_ = 3, *k*_2_ = 10) is the confidence factor, *N_d_* is the baseline noise, *C* is sample concentration, and *H* is the peak height of liquid chromatography.

(5)Matrix effect

A total of 1 g of *C. oleifera* saponins sample powder was accurately weighed, added to 50 mL of 80% methanol solution, soaked overnight, sonicated for 40 min, and centrifuged at 5000 r/min for 15 min. An amount of 2 mL of the supernatant was precisely transferred, dried under a vacuum oven at 60 °C, dissolved in methanol, and diluted to 5 mL. The extract was accurately transferred to a centrifuge tube containing 100 mg and centrifuged at 5000 r/min for 6 min. The supernatant was filtered through a 0.22 μm membrane, which was a matrix solution. The matrix effect was calculated using Formula (7).
(7)$$\mathrm{ME}=\left(\frac{S_{m}}{S_{s}}-1\right)\times 100\%$$

In the formula, *S_m_* is the slope of the working curve prepared by the matrix matching standard solution, and *S_s_* is the slope of the working curve prepared by the standard solution prepared by pure solvent. Additionally, |ME| < 20% is a weak matrix effect, which can be ignored without compensation measures; 20% ≤ |ME| ≤ 50% is a moderate matrix effect; |ME| > 50% is a strong matrix effect, and measures should be taken to compensate the matrix effect in this case.

##### Liquid Chromatography Detection Method

Based on the reference method, the study was carried out via liquid chromatography (1260 Infinity, Agilent Technologies, Santa Clara, CA, USA) under the following conditions [39].

We used an Eclipse XDB-C18 [40] column (4.6 × 250 mm, 5 μm); mobile phase: methanol—0.1% phosphoric acid; column temperature: 30 °C; detection wavelength: 267 nm; injection volume: 10 µL; flow rate: 0.8 mL/min; analysis time: 50 min. An elution gradient was set up with the methanol mobile phase increasing from 10% to 20% in 0–15 min, while the methanol mobile phase was increased from 20% to 40% between 15~30 min, from 40% to 45% between 30–40 min, and from 45% to 60% between 40–50 min. The results are shown in Figure 11.

##### Weight Detection

Alkali and acid hydrolysis of *C. oleifera* saponins was used to generate saponins. Based on the properties of hydrolysis, the hydrolyzed solution was injected into water to precipitate saponins of good quality [41].

#### 3.2.2. Alcohol Extraction of *C. oleifera* Saponins

Our research method of extracting *C. oleifera* saponins by methanol was improved and optimized on the basis of related purification technology [32]. Using an analytical balance, 15 g of degreasing camellia oil was weighed into a 250 mL round-bottomed flask, followed by the addition of 75% anhydrous methanol to obtain a liquid–solid ratio of 3.5:1. The mixture was condensed and refluxed at 60 °C for 180 min before filtering. The filter residue was washed with hot water, repeatedly. The filtrate was combined several times, heated, and stirred briefly in a constant-temperature water bath until about 10 mL of the filtrate remained. The remaining filtrate was left in a 54 °C vacuum drying box (DZ-2BCII, Tianjin Test Instrument Co., Ltd., Tianjin, China) to dry to constant weight. The crude *C. oleifera* saponins were obtained (crude *C. oleifera* saponins 1).

#### 3.2.3. Aqueous Two-Phase Extraction of *C. oleifera* Saponins

Our research method using an aqueous two-phase extraction was improved and optimized on the basis of the study of *C. oleifera* saponins purification [42]. Following the dissolution of 10 g ammonium sulfate in water, 11 g N-Propanol was added slowly while stirring. A certain amount of distilled water was added until the total weight of the system was 50 g, forming a two-phase system. The crude *C. oleifera* saponins 1 obtained after drying were added to a beaker and transferred to the aqueous two-phase system. It was stirred in a 30 °C water bath for 20 min, followed by transfer to a 250 mL liquid separation funnel for 20 min. The liquid *C. oleifera* saponins were removed and heated in a constant-temperature water bath at a certain temperature and stirred briefly until about 10 mL of the filtrate remained. The remaining liquid was dried in a 54 °C vacuum drying box to constant weight. The dried, solid *C. oleifera* saponins were designated as *C. oleifera* saponins 2. The extraction, purification, and detection of the *C. oleifera* saponins are shown in Figure 12.

#### 3.2.4. Statistics and Analyses

##### Statistical Analysis of *C. oleifera* Saponins Alcohol Extraction

In the present study, based on the previous single-factor experiment, the orthogonal array experiment was designed using SPSS 25.0., where the following four variables were analyzed in *C. oleifera* saponins alcohol extraction: extraction temperature (factor A), extraction time (factor B), methanol concentration (factor C), and liquid–solid ratio (factor D). These variables were identified to have larger effects on the yield of *C. oleifera* saponins from *C. oleifera* Abel. seed meal. The L9 (3^4^) matrix, which is an orthogonal array of four factors and three levels, was employed to assign the considered factors and levels, as shown in Table 7. Optimal conditions were obtained after the orthogonal experiments and subsequent data analysis, as shown in Table 5. Finally, the experiment was repeated under the optimal conditions in order to verify the data.

##### Statistical Analysis of *C. oleifera* Saponins Aqueous Two-Phase Extraction

Based on the previous single factor experiment, the orthogonal array experiment was designed using SPSS 25.0., where the following three variables were analyzed in the aqueous two-phase extraction process of *C. oleifera* saponins: extraction temperature (factor A), ammonium sulfate mass fraction (factor B), and propanol mass fraction (factor C). These variables were identified to have larger effects on the yield of *C. oleifera* saponins. The L9 (3^3^) matrix, which is an orthogonal array of three factors and three levels, was employed to assign the considered factors and levels, as shown in Table 8. Optimal conditions were obtained after the orthogonal experiments and subsequent data analysis, as shown in Table 6. Finally, the experiment was repeated under the optimal conditions in order to verify the data.

## 4. Conclusions

Compared with vanillin sulfuric acid colorimetry and gravimetric analysis, liquid chromatography was more accurate and sensitive, was simple to operate, and resulted in rapid analysis and strong separation. It eliminated the interference due to impurity and solvent peaks, and it was more suitable for the rapid quantitative detection of *C. oleifera* saponins. Orthogonal experiments were carried out on the basis of a single factor. The extraction and separation were optimized. The optimum technical conditions for the extraction of *C. oleifera* saponins by formaldehyde were obtained, and then the crude *C. oleifera* saponins were further separated and purified by an ammonium sulfate/propanol two-phase extraction method. The extraction and purification rate of *C. oleifera* saponins was 83.72%, and the average yield was 61.34%. Compared with the literature on the purification of *C. oleifera* saponins, the purity of *C. oleifera* saponins increased to a certain extent, indicating that this study provides an important reference for the rapid and efficient quantitative detection of *C. oleifera* saponins, and has potential application value in industrial extraction, separation, and purification.

## Figures and Tables

**Figure 1 molecules-28-02132-f001:**
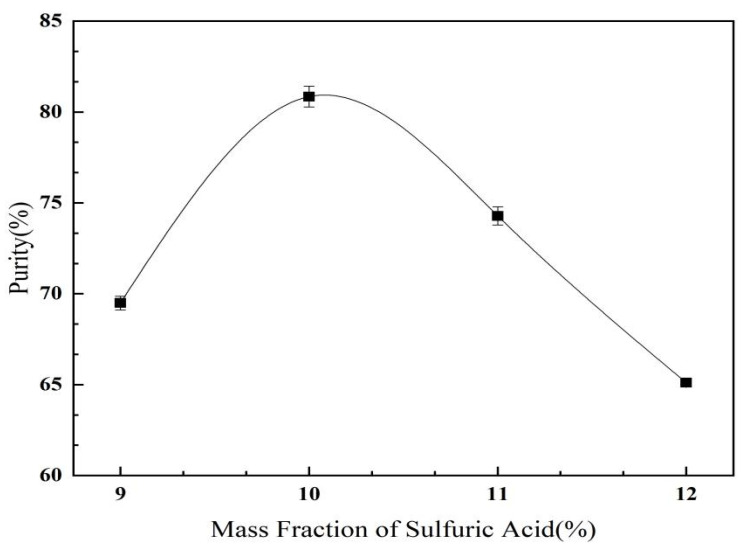
Standard curve of standard *C. oleifera* saponins by UV spectrophotometry.

**Figure 2 molecules-28-02132-f002:**
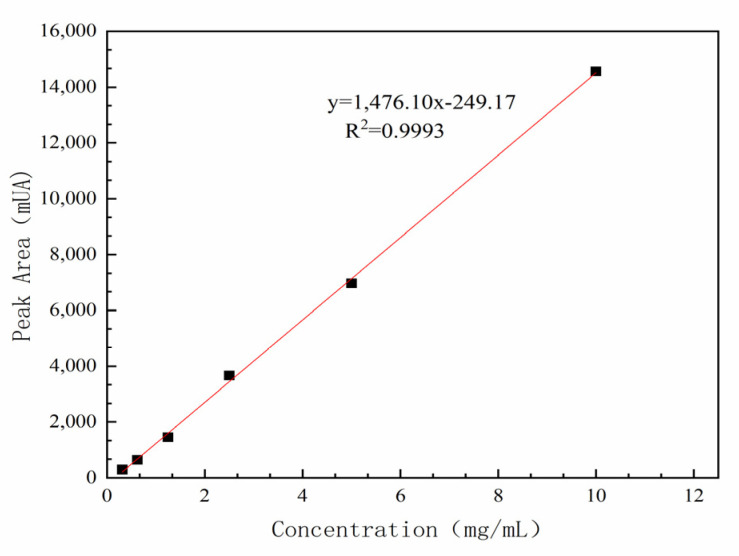
Standard curve of standard *C. oleifera* saponins by liquid chromatography.

**Figure 3 molecules-28-02132-f003:**
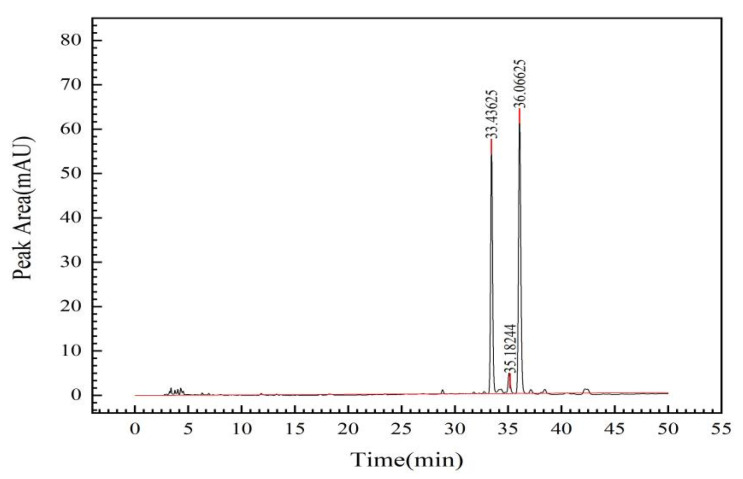
Liquid chromatography of *C. oleifera* saponins solid sample.

**Figure 4 molecules-28-02132-f004:**
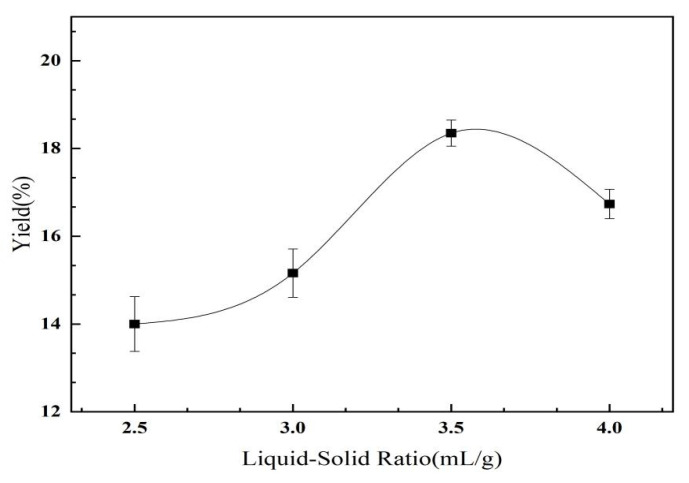
Influence of the extracted liquid–solid ratio on the yield of *C. oleifera* saponins (extraction temperature = 65 °C; methanol concentration = 75%; extraction time = 180 min).

**Figure 5 molecules-28-02132-f005:**
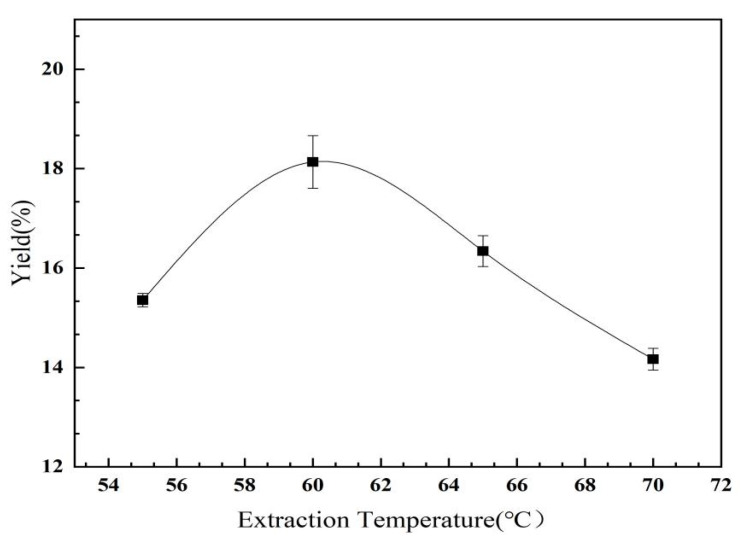
Influence of extraction temperature on yield of *C. oleifera* saponins (liquid–solid ratio = 3.5:1; methanol concentration = 75%; extraction time = 180 min).

**Figure 6 molecules-28-02132-f006:**
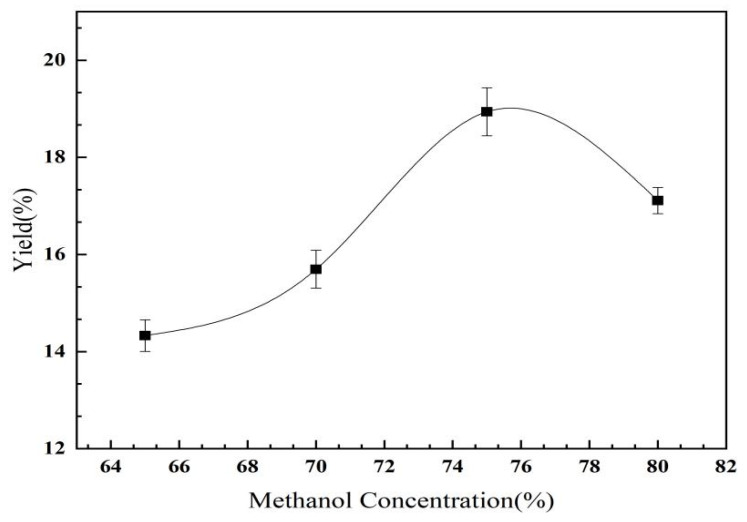
Influence of methanol concentration on the yield of *C. oleifera* saponins (liquid–solid ratio = 3.5:1; extraction temperature = 65 °C; extraction time = 180 min).

**Figure 7 molecules-28-02132-f007:**
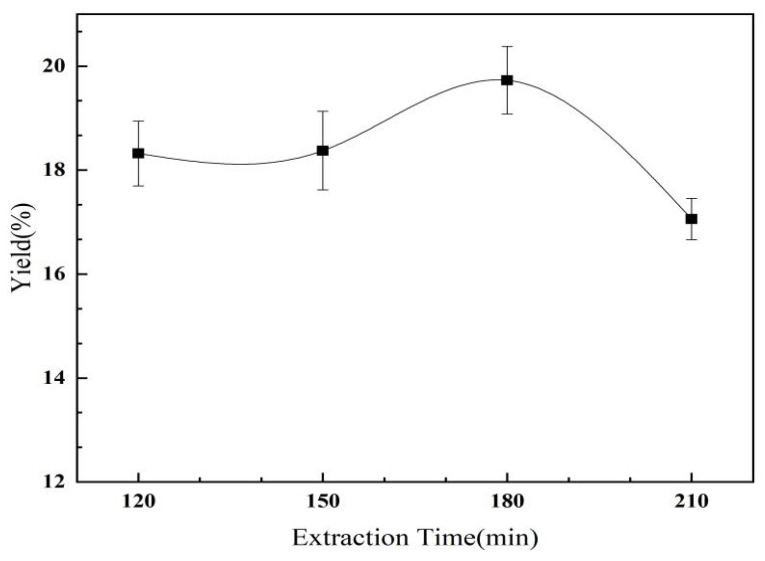
Influence of extraction time on yield of *C. oleifera* saponins (liquid–solid ratio = 3.5:1; extraction temperature = 65 °C; methanol concentration = 75%).

**Figure 8 molecules-28-02132-f008:**
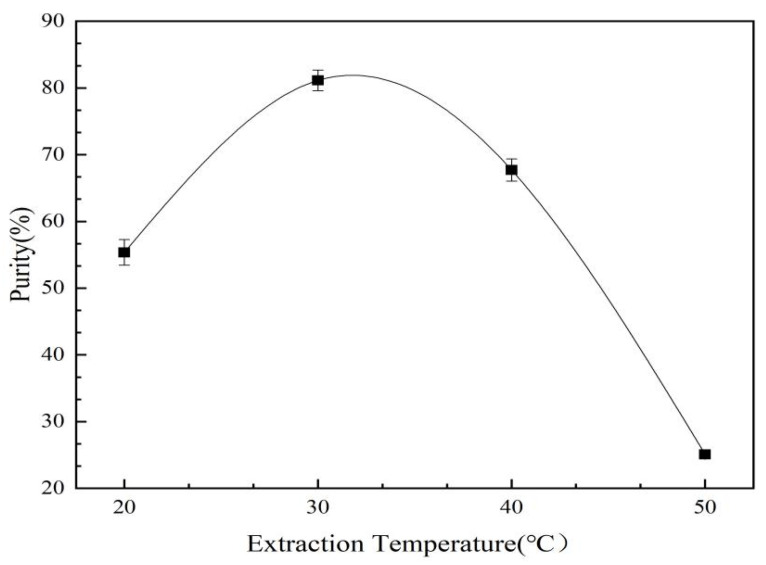
Effect of extraction temperature on the purity of *C. oleifera* saponins aqueous two-phase extraction (mass fraction of propanol = 11%; mass fraction of ammonium sulfate = 10%).

**Figure 9 molecules-28-02132-f009:**
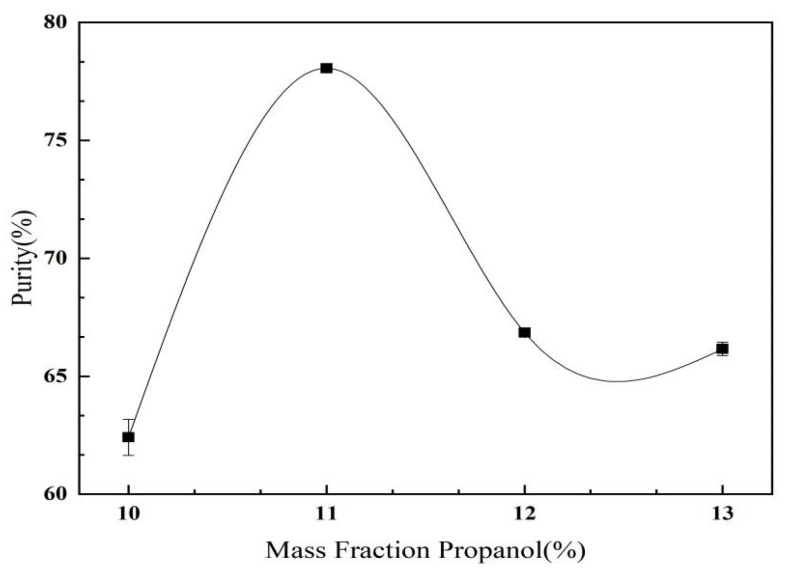
Effect of propanol mass fraction on the purity of *C. oleifera* saponins aqueous two-phase extraction (extraction temperature = 40 °C; mass fraction of ammonium sulfate = 10%).

**Figure 10 molecules-28-02132-f010:**
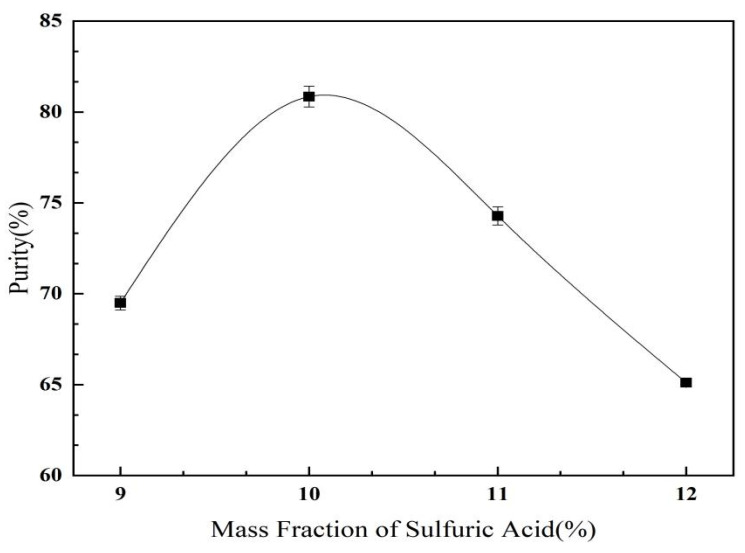
Effect of ammonium sulfate mass fraction on the purity of *C. oleifera* saponins aqueous two-phase extraction (extraction temperature = 40 °C; mass fraction of propanol = 11%).

**Figure 11 molecules-28-02132-f011:**
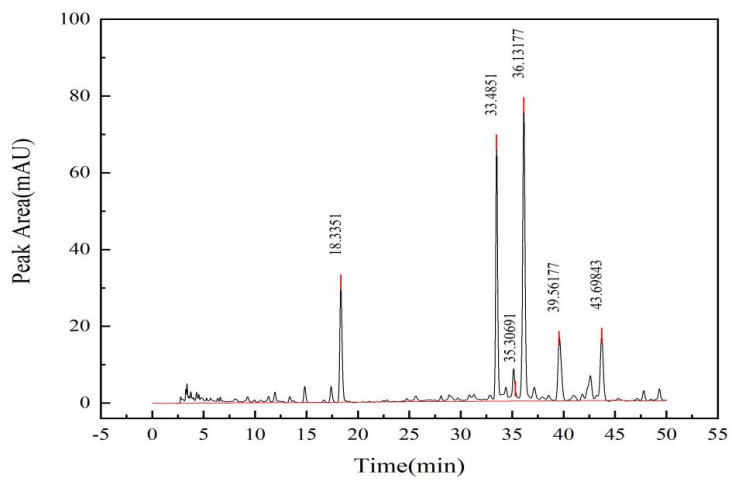
Detection results of liquid chromatography (column temperature = 30 °C; detection wavelength = 267 nm; injection volume = 10 µL; flow rate = 0.8 mL/min).

**Figure 12 molecules-28-02132-f012:**
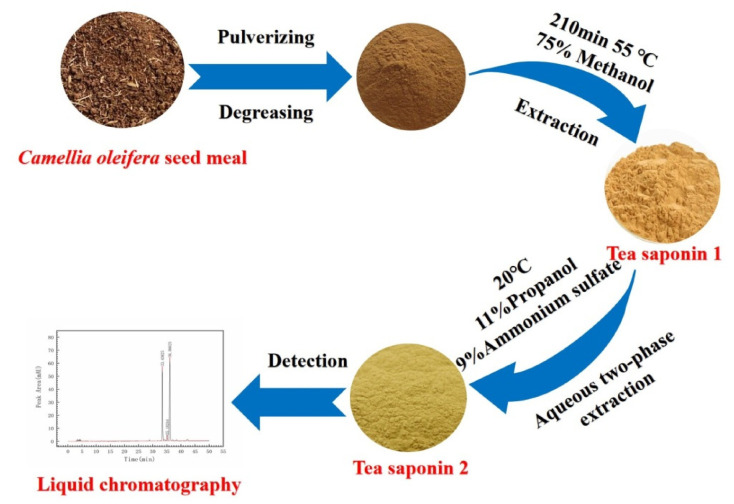
Extraction, purification, and detection of *C. oleifera* saponins.

**Table 1 molecules-28-02132-t001:** Liquid chromatography standard curve and corresponding peak area.

Serial Number	Concentration (mg/mL)	Injection Pressure (bar)	Peak Area (mAU)
1	10	109.36	14,556.5 0 ± 0.25
2	5	110.98	6970.40 ± 0.61
3	2.5	108.69	3657.0 0 ± 0.44
4	1.25	107.96	1448.58 ± 0.34
5	0.625	108.79	637.77 ± 0.48
6	0.3125	109.46	295.68 ± 0.52

**Table 2 molecules-28-02132-t002:** Precision parallel measurement.

Serial Number	Injection Pressure (bar)	Peak Area (mAU)
1	107.65	822.70 ± 0.84
2	106.96	824.30 ± 0.91
3	107.88	829.90 ± 0.74
4	105.75	830.00 ± 0.23
5	108.96	830.00 ± 0.43
6	108.96	830.10 ± 0.65

**Table 3 molecules-28-02132-t003:** Repeatability experimental data.

Serial Number	Sample Mass (mg)	Injection Pressure (bar)	Sum of Peak Areas (mAU)
1	10.01	107.96	1715.10 ± 0.82
2	9.96	105.69	1708.80 ± 0.64
3	10.01	108.46	1710.27 ± 0.72
4	9.99	107.65	1706.46 ± 0.44
5	9.99	107.88	1708.00 ± 0.52
6	10.00	106.44	1703.80 ± 0.48

**Table 4 molecules-28-02132-t004:** Recovery rate of standard addition.

Serial Number	Spiked Amount (mg)	Recovery Amount (mg)	Recovery Rate (%)
1	5.00	5.12	102.40 ± 0.33
2	5.00	4.89	97.80 ± 0.45
3	5.00	4.86	97.20 ± 0.38
4	10.00	10.13	101.30 ± 0.42
5	10.00	10.35	103.50 ± 0.22
6	10.00	9.98	99.80 ± 0.64
7	15.00	14.79	98.60 ± 0.55
8	15.00	15.68	104.50 ± 0.43
9	15.00	14.81	98.70 ± 0.28

**Table 5 molecules-28-02132-t005:** Orthogonal test results.

Experimental Number Factors	A	B	C	D	Yield/%
1	55 °C	150 min	70%	3:1	21.80 ± 0.21
2	55 °C	180 min	75%	3.5:1	22.21 ± 0.32
3	55 °C	210 min	80%	4:1	23.42 ± 0.14
4	60 °C	150 min	75%	4:1	24.07 ± 0.22
5	60 °C	180 min	80%	3:1	19.55 ± 0.24
6	60 °C	210 min	70%	3.5:1	17.46 ± 0.33
7	65 °C	150 min	80%	3.5:1	17.09 ± 0.25
8	65 °C	180 min	70%	4:1	21.13 ± 0.16
9	65 °C	210 min	75%	3:1	25.17 ± 0.20
K_1_	67.43	62.96	60.39	66.52	
K_2_	61.08	62.89	71.45	56.76	
K_3_	63.39	66.05	60.06	68.62	
k_1_	22.48	20.99	20.13	22.17	
k_2_	20.36	20.96	23.82	18.92	
k_3_	21.13	22.02	20.02	22.87	
Rj	2.12	1.06	3.8	3.95	

Abbreviations are as follows: A, extraction temperature; B, extraction time; C, methanol concentration; D, liquid to solid ratio; K, horizontal sum; k, mean value, K/3; Rj, range. The extraction mass was averaged three times.

**Table 6 molecules-28-02132-t006:** Orthogonal test result.

Experimental Number Factors	A	B	C	Purity/%
1	20 °C	9%	10%	76.62 ± 0.23
2	20 °C	10%	12%	57.21 ± 0.14
3	20 °C	11%	11%	75.32 ± 0.12
4	30 °C	9%	11%	83.80 ± 0.28
5	30 °C	10%	10%	64.57 ± 0.44
6	30 °C	11%	12%	55.42 ± 0.32
7	40 ℃	9%	12%	66.73 ± 0.36
8	40 ℃	10%	11%	65.43 ± 0.33
9	40 °C	11%	10%	73.10 ± 0.21
K_1_	209.15	227.15	214.29	
K_2_	203.79	187.21	224.55	
K_3_	205.26	203.84	179.36	
k_1_	69.72	75.72	71.43	
k_2_	67.93	62.40	74.85	
k_3_	68.42	67.95	59.79	
Rj	1.79	13.32	15.06	

Abbreviations are as follows: A, extraction temperature; B, ammonium sulfate mass fraction; C, propanol mass fraction; K, horizontal sum; k, mean value, K/3; Rj, range. The extraction mass was averaged three times.

**Table 7 molecules-28-02132-t007:** Orthogonal test factor selection level table.

Level	A	B	C	D
1	55 °C	150 min	70%	3:1
2	60 °C	180 min	75%	3.5:1
3	65 °C	210 min	80%	4:1

**Table 8 molecules-28-02132-t008:** Orthogonal test factor selection level table.

Level	A	B	C
1	20 °C	9%	10%
2	30 °C	10%	11%
3	40 °C	11%	12%

## Data Availability

Not applicable.

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
