# Peer review of "Quantitative Analysis of *Camellia oleifera* Seed Saponins and Aqueous Two-Phase Extraction and Separation"

_molecules, 2023, doi:10.3390/molecules28052132_

Round 1

Reviewer 1 Report

It a very interesting study but needs the following corrections:

1.      I suggest to change the title “Quantitative Analysis of Tea Saponins and Aqueous Two- Phase Extraction and Separation” to be Quantitative Analysis of Camellia oleifera seed Saponins and Aqueous Two- Phase Extraction and Separation”

2.      Add specie between thetea line 17.

3.      I recommend authors to replace the Tea word with its latin name  throughout the whole manuscript to be clear for readers which plant you investigated.

4.      In line 313, the authors mentioned “Camellia oleifera seed meal was provided by Jiangxi Zhongye Tea Technology Co., 313 Ltd.” the authors have to get Ethical approval from the company manufactured this product and add this approval number in the manuscript.

5.      Add space between numbers and units as in line 337 and many others.

6.      Add space between text and citations throughout the whole manuscript.

7.      Add suitable references to 3.2.3. Aqueous two-phase extraction of tea saponin and 3.2.2. Alcohol extraction of tea saponin sections.

8.      Add SD values to all obtained results especially in Tables 4-6

9.      add statistical analysis section at the end of material and methods part.

10.   The whole manuscript needs major grammar, typos and editing corrections by a native speaker.

Author Response

Referee: 1

1. I suggest to change the title “Quantitative Analysis of Tea Saponins and Aqueous Two- Phase Extraction and Separation” to be Quantitative Analysis of Camellia oleiferaseed Saponins and Aqueous Two- Phase Extraction and Separation”

Response: Thank you very much for your suggestion. We have modified these in revised version and marked them in red.

2. Add specie between the tea line 17.

Response: Thank you very much for your suggestion. We added space in revised version and marked in red.

3. I recommend authors to replace the Tea word with its latin name throughout the whole manuscript to be clear for readers which plant you investigated.

Response: Thank you very much for your suggestion. We have added its Latin name in revised version and marked in red.

4. In line 313, the authors mentioned “Camellia oleifera seed meal was provided by Jiangxi Zhongye Tea Technology Co., 313 Ltd.” the authors have to get Ethical approval from the company manufactured this product and add this approval number in the manuscript.

Response: Thank you very much for your suggestion. We have added this approval number and marked in red.

5. Add space between numbers and units as in line 337 and many others.

Response: Thank you very much for your suggestion. We have added spaces between numbers and units except % and °C and marked in red.

6. Add space between text and citations throughout the whole manuscript.

Response: Thank you very much for your suggestion. We have added spaces between text and citations throughout the whole manuscript.

7. Add suitable references to 3.2.3. Aqueous two-phase extraction of tea saponin and 3.2.2. Alcohol extraction of tea saponin sections.

Response: Thank you very much for your suggestion. We have added relevant references and marked them in red.

8. Add SD values to all obtained results especially in Tables 4-6

Response: Thank you very much for your suggestion. We have added SD values in all tables and marked them in red.

9. Add statistical analysis section at the end of material and methods part.

Response: Thank you very much for your suggestion. We have added statistical analysis sections at the end of material and methods part and marked them in red.

10. The whole manuscript needs major grammar, typos and editing corrections by a native speaker.

Response: Thank you very much for your suggestion. We have the manuscript polished by experts and marked the corrections in red. If there are problems with the grammar of the article, we will ask the professionals to modify it again.

Reviewer 2 Report

The manuscript entitled “Quantitative Analysis of Tea Saponins and Aqueous Two Phase Extraction and Separation” and authored by Zhu and colleagues deals with the validation of a detection method for saponins in Camellia oleifera samples by HPLC and UV/Vis methodologies.

The manuscript contains important information that seriously can contribute to the current state of the art.

My biggest concern about the manuscript reviewed is related to the small amount of data presented. Therefore, I strongly recommend changing the article type to Communication. Moreover, a number of changes are strongly recommended:

The email address of the authors should be included in the affiliations section, along with acronyms. The same acronyms should be those used in the contributions section.

The abstract should be a section consisting of a maximum of 200 words, in which the state of the art should be described. Next, a description of the methodologies and the main results obtained should be included. Finally, a concluding sentence and on future prospects should be added.

Some keywords should be changed. The utility of these terms is to facilitate the search of the article using common scientific search engines (PubMed, GoogleScholar, Scopus, etc.), which rely on the terms contained in title, abstract, and keywords. Consequently, using terms that are already in these sections as keywords is inappropriate. I strongly suggest that the repetitive keywords be changed before re-submission.

Figure captions should be seriously implemented. It should be more descriptive.

The Results and Discussion section is severely lacking in information previously published in the literature. The authors should implement this section because it is currently a simple discussion of the results obtained.

In an article related to the validation of an analytical method for saponins, I expected to find information related to the validation of the method, both in HPLC and UV/Vis. In particular, information related to limits of detection, quantification, matric effect, is completely missing. This information must necessarily be implemented in a manuscript targeting this type of analysis.

Author Response

Referee: 2

1. I strongly recommend changing the article type to Communication. Moreover, a number of changes are strongly recommended: The email address of the authors should be included in the affiliations section, along with acronyms. The same acronyms should be those used in the contributions section

Response: Thank you very much for your suggestion. The data content of our article meets the requirements of Article. If you insist, we can also modify it. We supplemented and improved the relevant information of the e-mail in accordance with the requirements and marked red.

2. The abstract should be a section consisting of a maximum of 200 words, in which the state of the art should be described. Next, a description of the methodologies and the main results obtained should be included. Finally, a concluding sentence and on future prospects should be added.

Response: Thank you very much for your suggestion. We modified the abstract according to the requirements and marked it in red.

3. Some keywords should be changed. The utility of these terms is to facilitate the search of the article using common scientific search engines (PubMed, GoogleScholar, Scopus, etc.), which rely on the terms contained in title, abstract, and keywords. Consequently, using terms that are already in these sections as keywords is inappropriate. I strongly suggest that the repetitive keywords be changed before re-submission.

Response: Thank you very much for your suggestion. We modified the keywords according to your suggestion and marked in red.

4. Figure captions should be seriously implemented. It should be more descriptive.

Response: Thank you very much for your suggestion. We modified the figure captions and marked in red.

5. The Results and Discussion section is severely lacking in information previously published in the literature. The authors should implement this section because it is currently a simple discussion of the results obtained.

Response: Thank you very much for your suggestion. We added this part in the conclusion and marked in red.

6. In an article related to the validation of an analytical method for saponins, I expected to find information related to the validation of the method, both in HPLC and UV/Vis. In particular, information related to limits of detection, quantification, matric effect, is completely missing. This information must necessarily be implemented in a manuscript targeting this type of analysis.

Response: Thank you very much for your suggestion. We added limits of detection, quantification and matric effect related data and computing formula, then marking in red.

Reviewer 3 Report

The manuscript was well-written and it is very interesting.  Three main remarks: i) the English should be revised by a native English speaker; ii) the plant material should be improved by adding more information (genotypes, varieties of Camellia oleifera?); iii) the conclusions should be written avoiding redundant information already discussed in the other sections and they should be focused only on the main results in relation to the aims of the study. A last suggestion: It could be useful to slightly reduce the Introduction section because of too much information. Some details are not interesting for the aims of this study.

Author Response

Referee: 3

1. The English should be revised by a native English speaker.

Response: Thank you very much for your suggestion. We modified the manuscript through experts and marked in red. If there are problems with the grammar of the article, we will ask the professionals to modify it again.

2. The plant material should be improved by adding more information (genotypes, varieties of Camellia oleifera?).

Response: Thank you very much for your suggestion. We added variety of Camellia oleifera and marked in red.

3. The conclusions should be written avoiding redundant information already discussed in the other sections and they should be focused only on the main results in relation to the aims of the study.

Response: Thank you very much for your suggestion. We modified and refined the conclusions in the end part and marked in red.

4. It could be useful to slightly reduce the Introduction section because of too much information. Some details are not interesting for the aims of this study.

Response: Thank you very much for your suggestion. We modified and refined the introduction section and marked in red.

Round 2

Reviewer 1 Report

The authors established most of the required corrections except:

1. In line 167 correct the citation.

2. After mentioning the plant Camellia oleifera you can use the abbreviated manner C. oleifera throughout the whole manuscript.

3. The whole manuscript needs major grammar, typos, and editing corrections by proof editing service

Author Response

1. In line 167 correct the citation.

Response: Thank you very much for your suggestion. We corrected the citation and marked in red.

2. After mentioning the plant Camellia oleifera you can use the abbreviated manner C. oleifera throughout the whole manuscript.

Response: Thank you very much for your suggestion. We used the abbreviated manner C. oleifera throughout the whole manuscript and marked it in red.

3. The whole manuscript needs major grammar, typos, and editing corrections by proof editing service

Response: Thank you very much for your suggestion. We use the language modification service provided by MDPI this time. We modified the manuscript through experts and marked in green.

Reviewer 2 Report

The authors properly attended to the indications; the manuscript is now suitable for the publication.

Author Response

Referee: 2

The authors properly attended to the indications; the manuscript is now suitable for the publication.

Response: Thank you for your valuable comments to make our manuscripts more scientific and complete.
